# PBP4 Is Likely Involved in Cell Division of the Longitudinally Dividing Bacterium *Candidatus* Thiosymbion Oneisti

**DOI:** 10.3390/antibiotics10030274

**Published:** 2021-03-09

**Authors:** Jinglan Wang, Laura Alvarez, Silvia Bulgheresi, Felipe Cava, Tanneke den Blaauwen

**Affiliations:** 1Bacterial Cell Biology & Physiology, Faculty of Science, Swammerdam Institute for Life Sciences, University of Amsterdam, Science Park 904, 1098 XH Amsterdam, The Netherlands; J.Wang3@uva.nl; 2Department of Molecular Biology, Umeå University, SE-901 87 Umeå, Sweden; laura.alvarez@umu.se (L.A.); felipe.cava@umu.se (F.C.); 3Environmental Cell Biology, University of Vienna, Althanstrasse 14 (UZA I), 1090 Vienna, Austria; bulghes3@univie.ac.at

**Keywords:** cell division, penicillin binding proteins, peptidoglycan, protein localization, symbiosis

## Abstract

Peptidoglycan (PG) is essential for bacterial survival and maintaining cell shape. The rod-shaped model bacterium *Escherichia coli* has a set of seven endopeptidases that remodel the PG during cell growth. The gamma proteobacterium *Candidatus* Thiosymbion oneisti is also rod-shaped and attaches to the cuticle of its nematode host by one pole. It widens and divides by longitudinal fission using the canonical proteins MreB and FtsZ. The PG layer of *Ca*. T. oneisti has an unusually high peptide cross-linkage of 67% but relatively short glycan chains with an average length of 12 disaccharides. Curiously, it has only two predicted endopeptidases, MepA and PBP4. Cellular localization of symbiont PBP4 by fluorescently labeled antibodies reveals its polar localization and its accumulation at the constriction sites, suggesting that PBP4 is involved in PG biosynthesis during septum formation. Isolated symbiont PBP4 protein shows a different selectivity for β-lactams compared to its homologue from *E. coli*. Bocillin-FL binding by PBP4 is activated by some β-lactams, suggesting the presence of an allosteric binding site. Overall, our data point to a role of PBP4 in PG cleavage during the longitudinal cell division and to a PG that might have been adapted to the symbiotic lifestyle.

## 1. Introduction

The growth mechanism of rod-shaped bacteria is one of best-studied, being exemplified by the Gram-negative and -positive model organisms *Escherichia coli* and *Bacillus subtilis*, respectively. *Candidatus* Thiosymbion oneisti is an uncultivable rod-shaped ectosymbiont of the nematode host *Laxus oneistus*. It is a sulfur-oxidizing species that belongs to the Gram-negative Gammaproteobacteria like *E*. *coli* [1]. Compared to other rods, the growth and division of *Ca.* T. oneisti do not follow the canonical mode. These bacteria grow in width instead of elongating and constrict along their long axis during cell division [2,3,4]. One pole (the proximal pole) of each cell is attached to the skin of the host to form a palisade-like structure perpendicular to the nematode’s surface (Figure 1). It is hypothesized that their longitudinal fission may facilitate symbiosis maintenance [3]. This reproductive mode of *Ca.* T. oneisti provides a chance to determine the flexibility of the mechanisms of rod-shape morphogenesis and develop a better understanding of the fundamentals of the fission process.

Usually, bacterial morphology is maintained by the cell wall, specifically by the peptidoglycan (PG) layer, providing structural strength and osmotic stress resistance. The PG layer is a covalently closed mesh made of glycan chains of β-linked *N*-acetylglucosamine and *N*-acetylmuramic acid cross-linked by short peptides [5]. Both cell growth and division require coordinated processes of cleavage of the existing PG layer and insertion of new units executed by two large protein complexes: elongasome and divisome [6]. The actin homologue protein MreB forms short polymers underneath the cytoplasmic membrane and recruits elongasome proteins to coordinate PG synthesis and turnover in rod-shaped bacteria [7]. The tubulin-like protein FtsZ assembles into filaments to form a ring at the prospective division site as the scaffold of divisome subunits [8]. Treadmilling of FtsZ polymers guide PG synthesis at the septum [9]. Within elongasome and divisome, a group of evolutionary related proteins, penicillin-binding proteins (PBPs), play crucial roles in PG polymerization and maturation. The active site of PBPs is the target of β-lactam antibiotics that mimic the D-alanyl-D-alanine dipeptide substrate of PBPs to form a covalent acyl–enzyme complex leading to PBP inactivation and growth inhibition. The number of different PBPs varies per bacterial species [10]. On the basis of their functions and catalytic activities, one can classify PBPs into three categories: classes A, B, and C. Bifunctional PBPs with both transglycosylase activity—catalyzing the elongation of glycan chains—and transpeptidase activity—crosslinking peptide bridges between glycan chains—are grouped into class A. They are often essential for cell growth and survival, for example, in *E. coli* either PBP1a or PBP1b should be present [11]. Class B PBPs, such as *E. coli* PBP2 and PBP3, are monofunctional transpeptidases. They are more conserved than class A PBPs and normally work in concert with a SEDS (shape, elongation, division, and sporulation) family protein as a complex. *E. coli* PBP2 and its SEDS partner RodA serve as key components of the elongasome to make the cylindrical wall PG and maintain rod shape. PBP3 and FtsW, a RodA homologue, have similar PG polymerase and crosslinking activity in the divisome for septal synthesis [12,13]. All other PBPs belong to class C, such as *E. coli* PBP4, PBP5, PBP6, and PBP7. They are generally not essential under laboratory conditions but involved in PG separation, maturation, and recycling as D,D-carboxypeptidase removing the last D-alanine of the stem pentapeptide side chain of PG subunits or as D,D-endopeptidase cutting peptide crosslinks between glycan strands [10,14].

In *Ca*. T. oneisti cell division, FtsZ filaments localize medially parallel to the cell’s long axis to form a discontinuous ellipse at the division site [3]. Remarkably, MreB polymers also align medially at the cell periphery [4], instead of perpendicular to the long axis at the cell periphery as in other rod-shaped bacteria [15]. The MreB filaments localize around the future division plane prior to the FtsZ ring and seem to be required for FtsZ ring formation [4]. D-Amino acid dipeptide probes can be incorporated into PG as the analogue of D-alanyl-d-alanine to detect new PG synthesis. The fluorescent labeling of ethynyl-D-alanyl-d-alanine (EDA-DA), a clickable probe, shows that new PG incorporation in live cells of *Ca*. T. oneisti mostly occurs during cell division. It starts at the two poles and proceeds centripetally with septum constriction. FtsZ colocalizes with these new PG insertion sites while the MreB inhibitor A22 blocks new PG synthesis [4]. These facts illustrate that the reproductive mode of *Ca*. T. oneisti deviates from the well-known ones.

In the genome draft of *Ca*. T. oneisti 6 putative PBP encoding genes are identified [16]. Compared to the rod-shaped Gram-negative model *E. coli*, the type and number of class A and B PBPs are identical. Only two class C PBPs, homologues of PBP4 and PBP6, are recognized, which is far less than the seven class C PBPs in *E. coli* [10]. In particular, the major carboxypeptidase PBP5, the most abundant PBP in *E. coli*, does not have a corresponding equivalent in the *Ca*. T. oneisti genome. *E. coli* PBP4 is encoded by the *dacB* gene and thought to be primarily a D, D-endopeptidase in vivo even with some carboxypeptidase activity in vitro. PBP6, the product of the *dacC* gene, is known as a D,D-carboxypeptidase with uncertain physiological role. An *E. coli* mutant lacking PBP4 does not exhibit growth defects or morphological aberrations [17]. Analysis of a set of mutants with multiple class C PBPs deleted demonstrates that the loss of PBP4 in a PBP5/PBP6 double mutant background induces severe shape deficiencies [18]. PBP4 localizes specifically at midcell during septation to participate in septal PG synthesis and its absence (Δ*dacB* strain) affects the timing of the divisome assembly [19]. The overexpression of PBP4 is toxic and leads to reduced numbers of PG crosslinks and pentapeptides side chains in vitro and in vivo [20]. The crystal structure of *E. coli* PBP4 contains three domains. Domain I formed by residues 1-80 and 294-477 is a conserved transpeptidase domain containing the active site serine 62. Domain II and III that are inserted into domain I have unique folds and unknown functions [21]. The C-terminal end of PBP4 was predicted to be very weakly associated with lipid head groups to make PBP4 not anchored into but associated with membrane [22]. In this study, we found that the PG composition of the symbiont deviated from thus far investigated gammaproteobacteria. The observed high crosslink percentage of the PG layer prompted us to investigate the cellular localization and in vitro activity of the PBP4 homologue of *Ca*. T. oneisti.

## 2. Results

### 2.1. Composition of Ca. T. Oneisti Cell Wall

The non-canonical division morphology of *Ca*. T. oneisti prompted us to investigate its murein structure and composition. In Table 1, a summary of the main features is given, and in Appendix A, the HPLC chromatogram and the assignment of the peaks can be found. The PG layer had a high percentage of crosslinks of 67% and the percentage of higher order crosslinks in the form of trimers, tetramers, and other crosslinked peaks was 11.5%. The percentage of crosslinks varies with bacterial species, and *Ca*. T. oneisti stands out with its high crosslink levels. Normally, Gram-positive bacteria have higher-level crosslinkage than Gram-negative species. The relative abundance of crosslinks in the PG of *E. coli* is approximately 35% and it contains only 6% higher order structures [23], whereas the PG of the Gram-positive *Staphylococcus aureus* has 85% crosslinked muropeptides [24]. The average glycan chain length of the symbiont is considerable shorter than that of *E. coli* with ±12.6 and ±21 disaccharide units, respectively [5]. As a comparison, the Alphaproteobacteria Gram-negative marine species *Caulobacter crescentus* has even shorter average chain length (±6 disaccharide units), and its crosslink percentage is around 52% [25].

### 2.2. PBP4 Homologue in Ca. T. Oneisti

To understand why the PG layer of *Ca*. T. oneisti has such a high-level of crosslinkage and how it is synthesized, we started to study PBPs that are responsible for PG construction. The genome encoded by *Ca*. T. oneisti hypothetical PBPs were identified by the Basic Local Alignment Search Tool (BLAST ) analysis on the basis of *E. coli* PBP sequences (Table 2). Among these PBPs, we noticed that only one D,D-endopeptidase appeared to be present in the *Ca*. T. oneisti genome, which was predicted to be a homologue of *E. coli* PBP4 (hereafter written as PBP4^EC^). On the basis of predicted molecular mass, we found that it is the sixth putative PBP in the *Ca*. T. oneisti genome (Table 2). Because of its clear homology to PBP4 of *E. coli*, it is likewise named PBP4 (hereafter written as PBP4^TO^) and not PBP6, as would have been custom. Given the PG crosslink cleavage by D,D-endopeptidase, we selected PBP4^TO^ for further investigation.

The protein sequence alignment between PBP4^TO^ and PBP4^EC^ is shown in Appendix A. PBP4^TO^ conserves the PBP active site signature SXXK (STLK, catalytic serine at positions 69), SXN (SNN), and KTG. The predicted signal peptide [26] at the N-terminus has 30 residues, being 9 residues longer than PBP4 of *E. coli*. A 3D model (Figure 2) of PBP4^TO^ based on amino acid threading from Phyre2 [27] shows a structure with three distinct domains similar to the crystal structure of PBP4^EC^ (Protein Data Bank (PDB) entry 2ex2). In this model, the predicted domain I is the relatively conserved penicillin binding domain with the active site signature mentioned before and carrying a large insertion folded into two other domains. Domain II and domain III are unique compared with other PBP structures, and their roles are still not clear. The smallest domain III consists of three β-sheets and has only 42 residues, which is almost 43% less than PBP4^EC^, whereas the sizes of the other two domains are similar. Although the real PBP^TO^ structure is not known, the striking reduction of PBP4 domain III is not unprecedented, and some other bacteria, such as *Mycobacterium tuberculosis*, even lack domain III in their type 4 PBPs. 

### 2.3. PBP4^TO^ Localized at the Poles and New PG Insertion Sites

To investigate the PBP4^TO^ localization, we fixed *Ca*. T. oneisti cells and immunolabeled them with a PBP4^EC^ antibody [19]. The antibody specificity and recognition of PBP4^TO^ were verified by Western blot in *Ca*. T. oneisti protein extracts (Figure 3a). Immunolabeling of PBP4^TO^ expressed from a plasmid in *E. coli* Δ*dacB* strain further confirmed the recognition of PBP4^TO^ by anti-PBP4^EC^ (Figure 3b,c).

In non-constricted *Ca*. T. oneisti cells, the PBP4^TO^ localized at both poles as small foci (Figure 4 and Appendix A). More PBP4^TO^ foci were visible at the onset of septation. In constricting cells, PBP4^TO^ localized at the new poles and at constriction sites (Figure 4 and Appendix A). The 90° rotated view of 3D structured illumination microscopy (3D SIM) images illustrate that the PBP4 foci were in the center of the division plane at the leading edges of the constriction (Figure 4b). As a comparison, MreB polymers localized at the circumference surface of the septation plane and the FtsZ elliptical ring assembled at the septation plane, while both PBP4^TO^ and MreB accumulated medially in the frontal view images (Figure 4b,c). The total fluorescence of PBP4^TO^ increased linearly with cell width, indicating that its concentration as function of the cell division cycle is constant (Figure 4f). To analyze the localization pattern changes during cell cycle, we plotted PBP4^TO^ fluorescence against the normalized cell length and width (Figure 4d,e, respectively). For distribution along the cell length, we classified 1258 cells into four groups on the basis of their cell width, and the PBP4^TO^ fluorescence of each group was plotted against the normalized cell long axis (Figure 4d). Fluorescence distribution of the thinnest (youngest) cell group presented high intensity at both ends of the cell’s long axis. These peaks flattened as cells widened and merged into one central peak in the thickest (deeply-constricting) cells. For the localization pattern along the cell width, we classified 1796 cells into four groups on the basis of increasing cell area, and PBP4^TO^ fluorescence of each group was plotted against the normalized cell short axis (Figure 4e). PBP4^TO^ was observed to accumulate in the central part of the short axis with increasing cell size.

A previous study demonstrated that the incorporation of new PG in *Ca*. T. oneisti (as detected by EDA-DA) appears to start at the cell poles in young cells and proceeds centripetally in dividing cells [4]. The PBP4^TO^ localization pattern resembles these new PG synthesis sites, indicating that PBP4^TO^ may be involved in new PG insertion. The difference between the PBP4^TO^ and the EDA-DA signal is that PBP4^TO^ also remains localized at the cell poles throughout the cell cycle. Septum synthesis in *Ca*. T. oneisti, including PG incorporation and membrane invagination, is mediated by the FtsZ ellipse throughout cell division. On the basis of these data, we speculate that PBP4^TO^ localizes at cell poles of newborn cells; when cell division starts, PBP4^TO^ appears in newly formed poles and also accumulates around constriction sites. PBP4^TO^ might be involved in new PG synthesis together with the divisome considering its localization is compatible with new PG incorporation sites. 

### 2.4. Active Site of PBP4^TO^

The cultivation of *Ca*. T. oneisti cells is still not possible no matter with or without their nematode host, nor is its gene editing. Therefore, only experiments in vitro are available to investigate the possible physiological function of PBP4^TO^. We cloned the *Ca*. T. oneisti *dacB* gene, inserted it into a pET302 vector for high expression in the cytoplasm, and purified the His-tagged protein from *E. coli* strain SF100 (Figure 5a). Bocillin-FL, a fluorescent penicillin V, is a commercial reagent that labels active PBPs covalently and allows their detection on SDS-PAGE gels. Purified PBP4^TO^ protein was detectable after Bocillin-FL labeling and a linear relationship between the fluorescence intensity and protein amount (Figure 5b) was observed as reported for other PBPs [41]. The active site serine 69 was replaced by alanine as non-functional control. However, although the S69A mutant lost most of its binding to Bocillin-FL compared with wild-type protein (Figure 5c), it was still able to bind some Bocillin-FL, suggesting the presence of an allosteric binding site. Some PBPs, from all classes, have been found to form dimers both in vivo and vitro [21,42,43]. The PBP4^EC^ forms dimers as isolated protein and its domain 3 is not essential for this dimerization [19]. PBP4^TO^ and PBP4^TO^ S69A also appears to form dimers and higher homo-oligomeric complexes in native PAGE gel (Figure 5d).

### 2.5. β-Lactam Antibiotics Binding Affinity

Inhibition ability of different β-lactams for different PBPs can be used to identify their discrete function [44]. For instance, aztreonam preferentially targets PBP3 of some Gram-negative bacteria, resulting in cell division interruption and cell wall breakage [45], but weakly binds to other PBPs. In this study, we selected seven β-lactams, which are deemed to be selective for various PBPs, in order to examine the affinity and selectivity of PBP4^TO^ in vitro (Figure 6). Among them, penicillin G had a similar affinity for PBP4^TO^ as for PBP4^EC^. PBP1b inhibitor cefsulodin, PBP2 inhibitor mecillinam, PBP4 inhibitor cephalexin, and the general PBP inhibitor ampicillin appeared to weakly bind PBP4^TO^ in the assay, although cephalexin and ampicillin are reported to have selectivity for PBP4^EC^ [44]. PBP3 inhibitor aztreonam does not show selective inhibition but surprisingly results in improved binding of Bocillin-FL at high concentration treatment. Meropenem, as a carbapenem antibiotic, is considered to have high affinity for PBP2 and PBP4 [46] and does present some affinity for PBP4^TO^, whereas Bocillin-FL intensity is enhanced with high-concentration treatment as well. The Bocillin-FL band intensity increase was observed in a previous study for *E. coli* PBP5 and PBP6 in live cells with high aztreonam concentration and oxacillin treatment without being related to protein concentration or accessibility but was only compound concentration-dependent [44].

### 2.6. Effects of Ca. T. Oneisti dacB Gene Overexpression in E. coli

No obvious growth defect or morphological change for *E. coli* Δ*dacB* strain has been described. However, the overexpression of PBP4^EC^ is toxic and results in cell lysis, possibly because of cell wall damage derived from PG crosslink cleavage by the protein [18]. To determine whether PBP4^TO^ can be functional in *E. coli*, we overproduced PBP4^TO^ from a low copy number plasmid vector with inducible p*Trc*-down promoter in the wild-type LMC500 strain and investigated the effects on cell growth and morphology. PBP4^TO^ expression did not affect growth rate while PBP4^EC^ expressed from the same plasmid backbone reduced cell growth significantly under the same conditions (Figure 7b). The shape defects observed upon overexpression of PBP4^EC^ were not seen in cells overexpressing PBP4^TO^ (Figure 7a). Although PBP4^TO^ and PBP4^EC^ were expressed from the same vector backbone, the level of the heterologous expression of PBP4^TO^ could be different. Therefore, a HA-tag was fused C-terminal of PBP4^TO/EC^ to assess the expression level. However, the Western blot analysis by anti-HA showed that PBP4^TO/EC^ were expressed at the same level (Appendix A). Using a stronger promoter p*Trc*, the expression of PBP4^TO^ caused a growth delay and reduced growth when it was induced by IPTG (Figure 7b), suggesting that PBP4^TO^ is less toxic for *E. coli* cells than PBP4^EC^. Nevertheless, membrane staining of live cells by MitoTracker Green FM demonstrates that both PBP4^TO^ and PBP4^EC^ overexpression from the same vector background leads to formation of cell membrane patches (Figure 7a), which indicates the phospholipid bilayer disorder. These membrane defects likely contribute to reduced cell growth, as shown in Figure 7b. 

## 3. Discussion

Gram-negative bacteria comprise 80–90% of total bacterial counts in the marine environment [47,48] but have been poorly investigated. The reason of their dominance in ocean habitats can be partially ascribed to the structural diversity of their cell wall, especially the LPS composition [49]. Herein, we investigated the PG composition and putative endopeptidase PBP4 of a Gram-negative symbiont living in tropical shallow water sediments. As a microbe with non-canonical growth mode, the PG of *Ca*. T. oneisti is higher crosslinked than other known Gram-negative species. This highly crosslinked PG structure perhaps may be an adaptation to its symbiotic lifestyle and unique mode of growing. The average chain length is ±12.6 disaccharide units, shorter than *E. coli* but not rare for Gram-negative bacteria. Both helical shaped *Helicobacter pylori* and crescent shaped *C. crescentus* have short glycan chain length (<10 units) [25,50]. Because *Ca*. T. oneisti is yet unculturable, the cells collected from its natural environment were sufficient to analyze its PG composition by mass spectrometry, but too limited to produce a UV absorbance profile for the PG. Hence, the result might not be completely comparable to previous data obtained from absorbance analysis at 204 nm on which most published data on the PG composition of common model organisms are based. However, using this method to analyze *Vibrio cholerae*’s PG [51] shows a degree of crosslinkage of 27.37% comparable to the 24.49% that was reported in [52] by the UV absorbance profile at 204 nm.

The PG layer is like a large mesh surrounding the cytoplasmic membrane that protects the cell and maintains its shape. During cell growth and division, the PG layer needs to be reshaped by cleavage of the existing structure to allow insertion of new PG units. Specifically, D,D-endopeptidases cleave the cross-link between side chains of old PG material to release PG fragments, which allow transglycosylases to incorporate new glycan chains and transpeptidases to synthesize new cross-links. Hence, class C PBPs, working as PG hydrolases, are necessary for cell growth and separation [53]. Most bacterial species have redundant endopeptidases and carboxypeptidases, such as the six confirmed endopeptidases and five carboxypeptidases in *E. coli*, for robustness of shape maintenance in different conditions [54]. However, in the genome of *Ca*. T. oneisti, only two putative homologues have been identified, corresponding to PBP4 and MepA of *E. coli*, and one putative carboxypeptidase, corresponding to *E. coli* PBP6. The observed number of amidases is also fewer than reported for other bacterial species. Although it cannot be excluded that yet some new type of unknown PG hydrolases exist in the *Ca*. T. oneisti genome, the PG hydrolases of *Ca*. T. oneisti are likely not as redundant as known for other species. The redundant hydrolases in *E. coli* form together with synthases and regulators of different PG synthesis complexes in response to the cell cycle state and changing environment [55]. The reason for less redundancy of endopeptidases and other hydrolases in *Ca*. T. oneisti may be related to its symbiotic lifestyle. The bacteria population coats the whole nematode body except its anterior region, which includes the head. The cell attachment is likely mediated by lectin proteins secreted from nematode glands [2]. These facts indicate that the growth rate of *Ca*. T. oneisti could be subject to host regulation. Moreover, association with the nematode provides symbionts with a relatively stable environment, which reduces external stresses [56]. This might allow a smaller adaptation range and therefore requires a less extended set of PG hydrolytic enzymes compared to free-living species.

Among the two putative identified endopeptidases in *Ca*. T. oneisti, MepA is homologous to a cation-dependent and penicillin-insensitive enzyme that is able to hydrolyze cross-linkages of dimer muropeptides in vitro but not murein under standard conditions in *E. coli* [57]. It could play a role at a specific growth stage or degrade PG fragments for recycling. The other putative endopeptidase PBP4^TO^ belongs to the class C PBP category, similar to PBP4^EC^, which is conserved in most bacteria except some Gram-positive cocci. On the basis of these facts, we speculate that PBP4^TO^ probably works as the dominant endopeptidase in PG construction during cell cycle in *Ca*. T. oneisti.

Isolated PBP4^TO^ protein can be labeled by Bocillin-FL, confirming its role as penicillin-binding protein as predicted by bioinformatics. Antibiotic affinity analysis illustrates that PBP4^TO^ has a different selectivity for β-lactams from PBP4^EC^ and appears to be most sensitive to penicillin G among the tested compounds. Ampicillin with only one amino group structural difference from penicillin G displays distinct selectivity, consistent with a previous study that antibiotics of similar structure do not always exhibit a similar affinity profile [44]. The Bocillin-FL signal of PBP4^TO^ increases with high-concentration treatment of meropenem and aztreonam. Similar band intensity increases were observed for PBP5/6^EC^ with high concentrations of aztreonam and oxacillin treatments [44]. The possible reason for the binding increase is the presence of an allosteric binding domain in these PBPs.

The crystal structure of some PBPs disclosed that an allosteric site exists outside of their transpeptidase domain [58,59]. PBPs with transpeptidase activity must accommodate two peptide side chains simultaneously to accomplish crosslinking of two glycan strands. This requires an extra binding site in addition to the common active site. It has been observed that binding of PG to the allosteric site could trigger a conformational change to open the standard active site of the transpeptidase domain for entry of the substrate, such as nascent PG or β-lactam antibiotic [59]. Allosteric domains have not been identified in class C PBPs. However, endopeptidase PBPs, which hydrolyze crosslinkage of peptide bridges, are also expected to serve two peptide chains and require two binding sites in theory. As a predicted endopeptidase, PBP4^TO^ might contain an allosteric binding site for β-lactams that have a low affinity for the penicillin-binding site of the protein. This allosteric binding then correlates with enhanced affinity of Bocillin-FL. The weak Bocillin-FL binding capability of mutant S69A also supports the presence of a second binding site.

The division of *Ca*. T. oneisti is FtsZ-dependent and its FtsZ ellipse localizes at the septum. FtsZ colocalizes with the new PG insertion pattern, which starts at the cell poles and proceeds centripetally with invagination [4]. Our data show that PBP4^TO^ localizes to the cell poles and constriction sites, where PG is inserted during invagination, suggesting that it participates in septal PG synthesis and works together with the divisome, maybe as a part of it. Intriguingly, in addition to its localization at the invagination region, the PBP4^TO^ is also present at the poles during the entire cell cycle. To organize the special longitudinal fission, *Ca*. T. oneisti cells possibly need mechanisms to achieve polar differentiation and correct septum positioning other than regular rod-shaped bacteria. In general, proteins known to function as polar landmarks have two possible strategies to localize at the cell poles. Either they can directly detect polar features (e.g., membrane curvature and specific membrane composition) or they interact with proteins that are already present at the cell poles [60]. Because PBP4^TO^ does not have structural similarity to proteins that can recognize polar features, it likely interacts with other polar proteins to maintain its localization.

The overexpression of PBP4^TO^ in *E. coli* causes slight growth defects, suggesting that PBP4^TO^ can be partially active in *E. coli*. Compared to PBP4^EC^, PBP4^TO^ overexpression is less toxic to *E. coli* cells. The PBP4^EC^ localizes in between PG layer and outer membrane. Domain III of PBP4^EC^ is essential for its midcell localization during cell division and for the interaction with the outer membrane lipoprotein NlpI [19,61]. PBP4^TO^ has a very small domain III, which likely does not interact with the partners of PBP4^EC^. Possibly, PBP4^EC^ activity is directed by its partners through its domain III, and overexpression allows its unrestricted and therefore toxic activity. We hypothesize that, in the absence of partners, the basal activity of PBP4^TO^ is very low and, therefore, not sufficient to have an effect on the PG layer of *E. coli*, unless it is considerably overexpressed. Possibly PBP4^TO^ needs to be activated by partners or it recognizes a slightly different PG structure that is not abundant in *E. coli.*

In conclusion, the PG layer of longitudinal dividing *Ca*. T. oneisti is highly cross-linked and consists of short glycan chains length with 12 disaccharides on average. PBP4^TO^ localizes polarly and at constriction-sites, suggesting that PBP4^TO^ is involved in PG synthesis during cell division. In vitro, it presents different antibiotic affinity from its homologue PBP4^EC^ and is less damaging to *E. coli* cells when overexpressed.

## 4. Materials and Methods 

### 4.1. Bacteria Strains and Plasmids

Symbiont *Ca*. T. oneisti were collected from a sand bar of Carrie Bow Cay, Belize (16 48′11.01″ N, 88 4′54.42″ W). Bacteria together with their host nematode were fixed by methanol and transported to Vienna deep-frozen [4]. The *E. coli* strains and plasmids used in this study are listed in Appendix A. The *dacB* gene, as described in Table 2, was amplified from *Ca*. T. oneisti DNA by PCR reaction. The genome sequence draft is available at http://rast.nmpdr.org/rast.cgi (accessed on 15 December 2020) for primer design. Primers are listed in Appendix A. Methanol fixed nematodes were rehydrated in Phosphate Buffered Saline (PBS: pH 7.4; 137 mM NaCl, 2.7 mM KCl, 10 mM Na_2_HPO_4_, 1.8 mM NaH_2_PO_4_), and symbiont bacteria were detached in a sonication bath for 1 min sonication. One microliter bacterial suspension was used as template for 25 μL PCR reaction and PCR condition was as follows: 98 °C for 10 min, followed by 30 cycles at 98 °C for 30 s, 57 °C for 45 s, 72 °C for 1 min, followed by a final elongation step at 72 °C for 10 min. The *dacB* gene is a 1245 nucleotide-long fragment encoding a 414 amino acid-long protein.

### 4.2. Expression and Purification of PBP4^TO^


PBP4^TO^ and PBP4^TO^ S69A proteins were isolated from strain SF100. Cells expressing PBP4^TO^ or S69A from plasmid pJW06/pJW07 were grown in rich medium (10 g Tryptone (Bactolaboratories, Australia), 5 g yeast extract (Duchefa, Amsterdam, The Netherlands) and 5 g NaCl (Merck, Kenilworth, NJ, USA) per liter) with 15 µg/mL chloramphenicol at 28 °C and protein expression was induced by 0.5 mM IPTG for 6 h. Harvested cells were resuspended in 10 mM Tris buffer (10 mM Tris HCl (pH 7.4), 50 mM NaCl, and 10 mM MgCl_2_) and broken by French Press under pressure of 800 psi. Cell extracts were separated by ultracentrifugation 200,000× *g*. Supernatant was loaded onto a HisTrap HP prepacked column using an AKTA system (GE Healthcare) that was equilibrated with Tris buffer before loading of the samples. The column was washed with Tris buffer for 10 column volumes. Protein was eluted using a gradient of 5–100 mM imidazole in Tris buffer and 1 mL fractions were collected. Purified protein was confirmed by 10% SDS-PAGE.

### 4.3. Immunostaining 

Methanol-fixed *Ca*. T. oneisti bacteria attached on their nematode were rehydrated in PBS (pH 7.4) and washed once. Subsequently, bacteria were detached from nematode in a microcentrifuge tube by sonication bath for 1 min and permeabilized with lysozyme (1% *w*/*v* solution in PBS with 5mM ethylenediaminetetraacetic acid) for 10 min at room temperature. Polyclonal *E. coli* anti-PBP4 was incubated with *E. coli* Δ*dacB* strain to adsorb IgG with nonspecific binding for antibody pre-purification before immunostaining of *Ca*. T. oneisti. Blocking was performed in 0.5% blocking reagents in PBS (Boehringer) at 37 °C for 1 h. *Ca*. T. oneisti cells were incubated with 1:200 diluted pre-purified anti-PBP4 overnight at 4 °C and washed 3 times in PBS with 0.1% Tween after incubation. Secondary antibody, donkey anti-rabbit conjugated to Cy3 or Alexa488 (Jackson ImmunoResearch, USA), was diluted 1:500 in blocking solution and incubated with cells at 37 °C for 1 h. Unbound antibody was removed by washing 3 times in PBS with 0.1% Tween. Final sample was resuspended in 10 μL 1xPBS.

Exponentially growing *E. coli* cells were fixed with 2.8% formaldehyde and 0.04% glutaraldehyde while shaking in TY medium for 15 min. Cells were permeabilized and immunolabelled with pre-purified antibodies as described [62].

### 4.4. Microscopy and Image Analysis 

For standard fluorescence microscopy imaging, immunolabeled cells were immobilized on 1% agarose in PBS [63] and photographed with a CoolSnap *fx* (photometrics) CCD camera mounted on an Olympus BX-60 microscope with a 100X/ *N.A.* 1.35 oil objective (UPLANFI). Images were taken by the program ImageJ [64] with MicroManager.

For 3D structured illumination microscopy (3D SIM) imaging, cell suspensions were mounted on slides as described above and photographed using a Nikon Eclipse Ti N-SIM E microscope setup equipped with a CFI SR Apochromat TIRF 100X oil objective (*N.A.*1.49), a LU-N3-SIM laser unit, an Orca-flash 4.0 SCMOS camera (Hamamatsu Photonics K. K.), and NIS elements Ar software.

Localization pattern was analyzed using public domain program ImageJ [64] in combination with plugin ObjectJ and a modified version of Coli-inspector [4,65].

### 4.5. Bocillin-FL Binding Assay

The β-lactam antibiotics affinity of PBP4^TO^ was tested using a fluorescent penicillin derivative Bocillin-FL (Thermo-Fisher) [44]. One microgram of PBP4^TO^ in 10 mM Tris-HCl buffer (pH 7.5) with 50 mM NaCl and 10 mM MgCl_2_ was incubated with a gradient of antibiotics at 28 °C for 20 min. Then, 15 μg/mL Bocillin-FL was added to the reaction mixture and further incubated at 28 °C for 30 min. The reactions were stopped by addition of 5x SDS-PAGE sample buffer. Samples were loaded onto a 12% SDS-PAGE gel and proteins were separated by electrophoresis for 2 h at 120 V. The Bocillin-FL signal was detected with an Odyssey Fc imaging system (LI-COR Biosciences) at 600 nm.

### 4.6. Peptidoglycan Purification and Analysis 

PG samples were analyzed as described previously with some modifications [66,67]. Briefly, cells were pelleted and resuspended in 400 µL H_2_O with 5% SDS and boiled for 1 h. Sacculi were washed twice with MilliQ water by ultracentrifugation (110,000 rpm, 30 min, 20 °C) and finally resuspended in 40 μL H_2_O. The samples were treated with muramidase (100 μg/mL) for 16 h at 37 °C. Muramidase digestion was stopped by boiling for 15 min and coagulated proteins were removed by centrifugation (15 min, 14,000 rpm). For sample reduction, the pH of the supernatant was adjusted to pH 8.5–9.0 with sodium borate buffer, and sodium borohydride was added to a final concentration of 10 mg/mL. After incubating for 30 min at room temperature, pH was finally adjusted to 3.5 with orthophosphoric acid.

Detection and characterization of muropeptides by LC–MS was performed on an UPLC ™ system interfaced with a Xevo G2/XS Q-TOF mass spectrometer (Waters Corporation, Milford, MA, USA) equipped with an ACQUITY UPLC BEH C18 Column (130Å, 1.7 μm, 2.1 mm × 150 mm (Waters, USA)). Muropeptides were separated at 45 °C using a linear gradient from buffer A (formic acid 0.1% in water) to buffer B (formic acid 0.1% in acetonitrile) in an 18-min run, with a 0.25 mL/min flow. The QTOF–MS instrument was operated in positive ionization mode. Detection of muropeptides was performed by MS^E^ to allow for the acquisition of precursor and product ion data simultaneously, using the following parameters: capillary voltage at 3.0 kV, source temperature to 120 °C, desolvation temperature to 350 °C, sample cone voltage to 40 V, cone gas flow 100 L h^−1^, and desolvation gas flow 500 L h^−1^. Mass spectra were acquired at a speed of 0.25 s/scan. The scan was in a range of *m/z* 100–2000. Data acquisition and processing was performed using UNIFI software package (Waters Corp.). An in-house compound library built in UNIFI was used for detection and identification of muropeptides. Subsequent identification and confirmation of each muropeptide was performed by comparison of the retention-times and mass spectrometric data to known samples. Quantification was performed by integrating peak areas from extracted ion chromatograms (EICs) of the corresponding *m/z* value of each muropeptide and normalized to their molar ratio.

Main PG features were calculated as follows: percentage of monomers, dimers, trimers, and tetramers was calculated by adding the relative molar abundances of the different oligomers; overall crosslink was calculated as Dimers + (Trimers × 2) + (Tetramers × 3); percentage of anhydro muropeptides was calculated by adding the relative molar abundances of the different anhydro species; and average glycan chain length was calculated by dividing 100 by the percentage of anhydro muropeptides.

## Figures and Tables

**Figure 1 antibiotics-10-00274-f001:**
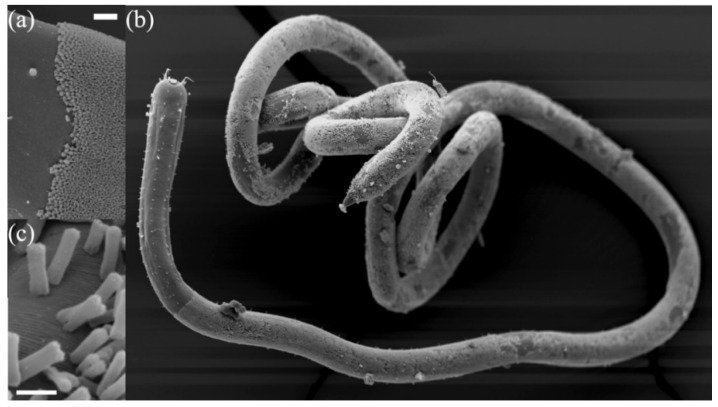
Scanning electron microscope images of *Candidatus* Thiosymbion oneisti and its nematode host *Laxus oneistus*. (**a**) The transition of the nematode’s bare skin and the start of the *Ca*. T. oneisti coat. The scale bar equals 20 μm. (**b**) *L. oneistus* covered by one single layer of *Ca*. T. oneisti. (**c**) Close up of dividing cells on the surface of the nematode. The scale bar equals 2 μm.

**Figure 2 antibiotics-10-00274-f002:**
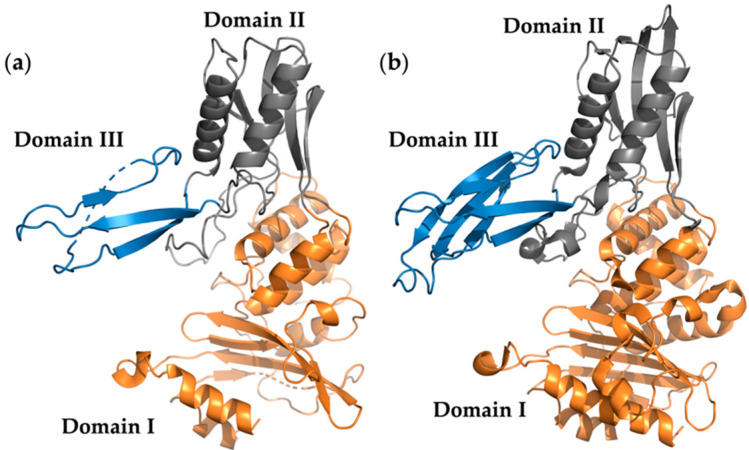
Structure model of penicillin-binding protein 4 (PBP4). (**a**) *Ca*. T. oneisti PBP4 structure model. Domains I, II, and III are colored as orange, gray, and blue, respectively. (**b**) *E. coli* PBP4 crystal structure (PDB entry 2EX2). Domains I, II, and III are colored as in (**a**).

**Figure 3 antibiotics-10-00274-f003:**
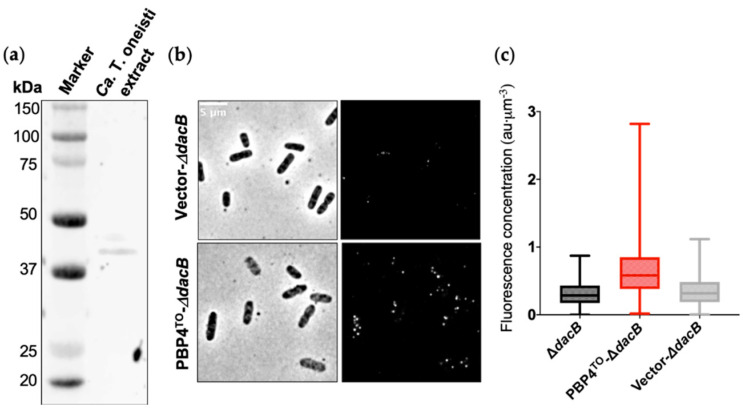
*E. coli* PBP4 polyclonal antibody recognize *Ca*. T. oneisti PBP4. (**a**) Western blot of *Ca*. T. oneisti extract developed with *E. coli* PBP4 polyclonal antibody. The predicted molecular weight of PBP4^TO^ was 45.2 kDa. (**b**) Phase contrast (**left**) and fluorescence (**right**) images of PBP4^TO^ immunolabeled in *E. coli* Δ*dacB* strain that expressed PBP4^TO^ from plasmid pJW05 induced with 50 μM isopropyl β-d-1-thiogalactopyranoside (IPTG) for 4 h grown in rich medium at 28 °C. Δ*dacB* strain with empty vector (upper images) showed almost no signal and Δ*dacB* with PBP4^TO^ expression from plasmid (lower images) showed signal distributed evenly around the cell periphery. (**c**) Fluorescence concentration (a.u. per µm^3^) analysis of images shown in (**b**). Number of cells analyzed for Δ*dacB*, Δ*dacB* with expression of PBP4^TO^, and Δ*dacB* with empty vector were 614, 1062, and 713, respectively.

**Figure 4 antibiotics-10-00274-f004:**
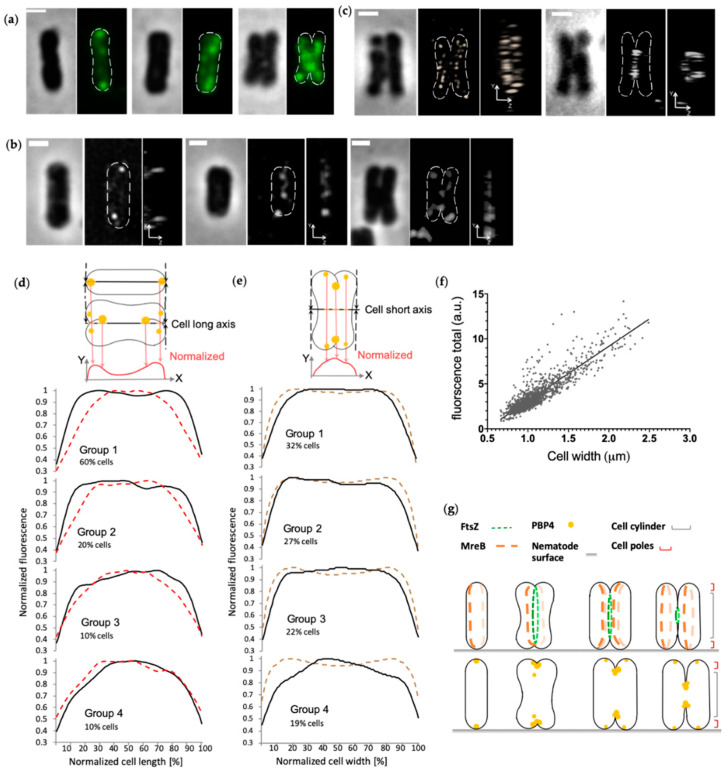
PBP4 localization in *Ca*. T. oneisti. (**a**) Phase contrast and fluorescence images of three representative *Ca*. T. oneisti cells of PBP4 immunolabeling. The scale bar is 1 μm. (**b**) 3D structured illumination microscopy (3D SIM) images of PBP4^TO^ localization. Three representative *Ca*. T. oneisti cells immunolabeled with PBP4^TO^ are arranged from thinnest (left) to thickest (right). The scale bar is 1 μm. A frontal view and corresponding 90° rotation view are shown for each cell. (**c**) A dividing *Ca*. T. oneisti cell immunolabeled with anti-MreB (left) and anti-FtsZ (right). The scale bar is 1 μm. A frontal view and corresponding 90° rotation view are shown. (**d**) Normalized fluorescence signal distribution of PBP4^TO^ (black lines) and MreB (red dashed lines) plotted against normalized cell length. Labeled cells were divided into four groups on the basis of their cell width, and are shown from the upper to the lower. Group 1 contains the thinnest cells with 0.66–1.03 μm cell width for PBP4^TO^ population and 0.6–0.96 μm for MreB. These are for both 60% of the total population (1258 and 1118 cells for PBP4^TO^ and MreB, respectively). Group 2 (20% of total population) contains 1.03–1.14 μm cells for PBP4^TO^ and 0.96–1.09 μm cells for MreB. Group 3 (10% of total population) contains 1.14–1.27 μm cells for PBP4^TO^ and 1.09–1.20 μm cells for MreB. Group 4, the thickest cells, is 10% of the total population and contains 1.27–1.50 μm cells for PBP4^TO^ and 1.20–1.50 cells for MreB. (**e**) Normalized fluorescence signal distribution of PBP4^TO^ (black lines) and corresponding cell diameter based on the corresponding phase contrast images (brown dashed line) plotted against normalized cell width. Labeled cells were divided into four groups on the basis of increasing cell area. Groups 1, 2, 3, and 4 contained 32%, 27%, 22%, and 19% of the cells in the population sorted according to their cell area with smallest on top and the largest cells at the bottom of the graphs, respectively. (**f**) Total PBP4^TO^ fluorescence plotted against cell width. The fluorescence shows a linear increase with cell width. (**g**) Schematic representation of the localization patterns of FtsZ and MreB as reported [3,4] and the localization pattern of PBP4^TO^ in the corresponding cell division cycle stages.

**Figure 5 antibiotics-10-00274-f005:**
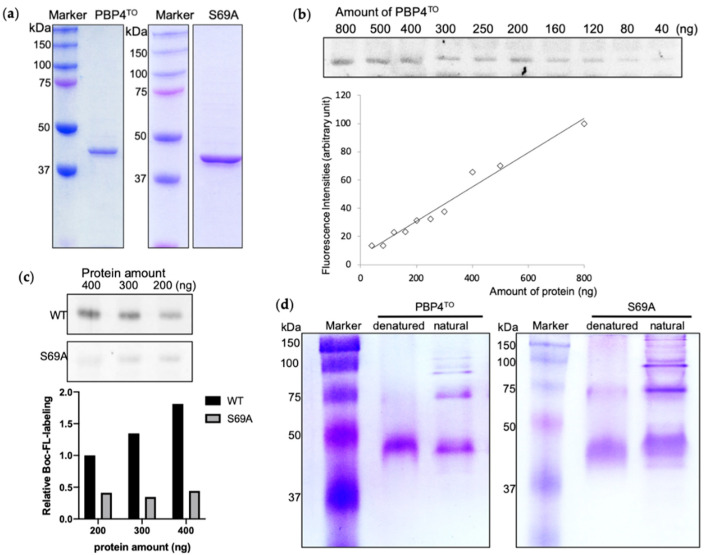
Purified PBP4^TO^ and PBP4^TO^ S69A (**a**) SDS-PAGE analysis of purified PBP4^TO^ and PBP4^TO^ S69A. (**b**) The binding of Bocillin-FL to PBP4^TO^. Bocillin-FL labeling signal showed a linear relationship with the amount of PBP4^TO^ protein. (**c**) Comparison of Bocillin-FL binding to PBP4^TO^ S69A. PBP4^TO^ S69A lost most of Bocillin-FL binding capacity compared to wild-type PBP4^TO^. (**d**) Native gel analysis of purified PBP4^TO^ and PBP4^TO^ S69A.

**Figure 6 antibiotics-10-00274-f006:**
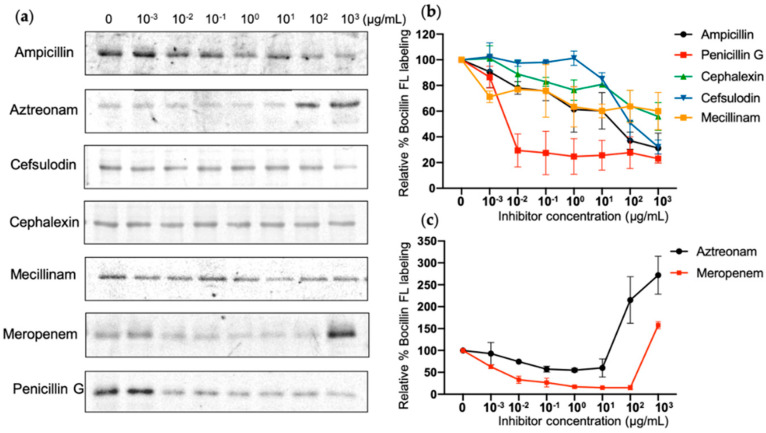
β-Lactam binding affinity of PBP4^TO^. (**a**) Representative SDS-PAGE gel images for antibiotic gradient titration of PBP4^TO^. The same amount of PBP4^TO^ was treated with various concentrations of antibiotics and labeled subsequently by Bocillin-FL. (**b**,**c**) Image quantification for antibiotic gradient with standard deviations (*n* = 2).

**Figure 7 antibiotics-10-00274-f007:**
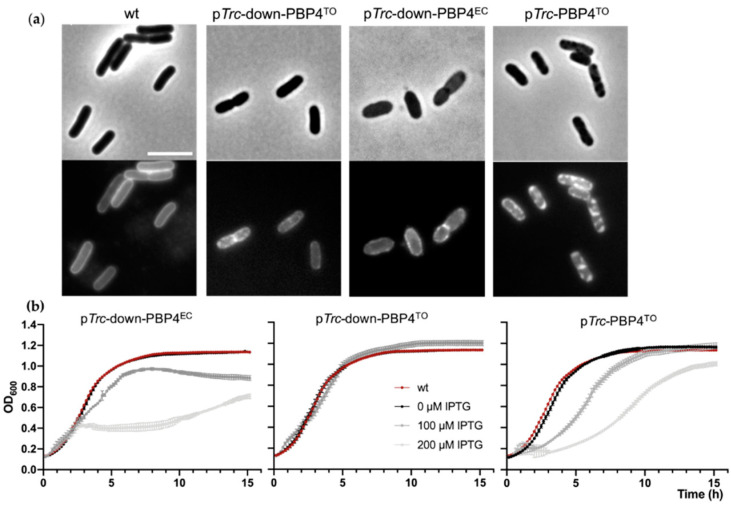
Mitotracker Green FM revealed membrane defects in *E. coli* LMC500 cells overexpressing PBP4^TO/EC^. *E. coli* cells were cultured in TY medium at 28 °C. (**a**) PBP4^TO/EC^ overexpression resulted in membrane defect of *E. coli* cells. Phase contrast (upper) and fluorescence (lower) images of wild type cells with PBP^TO/EC^ overexpression by 50 μM IPTG. Cell membrane was stained with MitoTracker Green FM. The scale bar equals 5 μm. (**b**) Growth curves of wild-type *E. coli* with PBP4^EC^ overexpression under weak promoter p*Trc*-down from plasmid pXL133 (left), PBP4^TO^ overexpression under control of the weak promoter p*Trc*-down from plasmid pJW05 (middle), and PBP4^TO^ overexpression under control of the strong promoter p*Trc* from plasmid pJW06 (right) induced with gradient IPTG. The values are mean ± SD from three repeats.

**Table 1 antibiotics-10-00274-t001:** Peptidoglycan composition of *Ca*. T. oneisti.

Features	Molar Percentage (%)
Monomers	45.16
Dimers	43.29
Trimers	10.54
Tetramers	1.01
Crosslinks	67.40
Anhydro muropeptide	7.95
**Features**	**Disaccharide subunits**
Chain length	12.58

**Table 2 antibiotics-10-00274-t002:** Set of PBPs from *Ca*. T. oneisti compared with *Escherichia coli.*

PBP^TO^	Gene^TO^	PBP^EC^	Putative Activity	Class	Reference
PBP1A	*mrcA*	PBP1A	TG and TP	A	[28,29]
PBP1B	*mrcB*	PBP1B	TG and TP	A	[28,29]
N/A	N/A	PBP1C	TG	A	[30]
PBP2	*mrdA*	PBP2	TP of elongasome	B	[31,32]
PBP3	*ftsI*	PBP3	TP of divisome	B	[12,31]
PBP3b ^1^	*pbp3b*	N/A	TP	B	N/A
PBP4	*dacB*	PBP4	DD-EPase	C type-4	[33]
N/A	N/A	PBP5	DD-CPase	C type-5	[34,35]
PBP5	*dacC*	PBP6	DD-CPase	C type-5	[36]
N/A	N/A	PBP6b	DD-CPase	C type-5	[37]
N/A	N/A	PBP7	DD-EPase	C type-7	[38]
N/A	N/A	PBP4b	DD-CPase	C type-AmpH	[39]
N/A	N/A	AmpH	DD-EPase and DD-CPase	C type-AmpH	[40]

TG = transglycosylase; TP = transpeptidase; DD-EPase = D-alanyl-d-alanine endopeptidase; DD-CPase = D-alanyl-d-alanine carboxypeptidase; N/A, no corresponding homologue or reference; ^1^ PBP3b is an unknown class B PBP in *Ca*. T. oneisti genome with predicted size 58kDa, smaller than PBP3 of *Ca*. T. oneisti.

## Data Availability

The data that support the findings of this study are available from the corresponding author upon reasonable request.

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
