# Peer review of "PBP4 Is Likely Involved in Cell Division of the Longitudinally Dividing Bacterium Candidatus Thiosymbion Oneisti"

_antibiotics, 2021, doi:10.3390/antibiotics10030274_

Round 1
Reviewer 1 Report
The manuscript is the third installment in a series by the same authors investigating growth and division in the longitudinally dividing bacterium Candidatus Thiosymbion oneisti. Briefly, the authors used LC-MS to determine cell wall composition in Ca. T. oneisti, which revealed a highly cross-linked wall composed of shorter glycans in line with other marine bacteria (e.g., Caulobacter crescentus). A BLAST analysis of the Ca. T. oneisti genome revealed a reduced complement of PBPs (when compared to E. coli) and the presence of one probable D,D-endopeptidase, PBP4TO. Follow-up studies show that PBP4TO is polar localizing, but localizes less convincingly to sites of division. Additional studies confirmed that PBP4TO is a bona fide penicillin binding protein but the authors did not assess whether PBP4TO exhibits its expected cell wall degrading activity.
Overall, the manuscript gives a first look at the structure of the cell wall from a bacterium with a somewhat unusual mode of growth. Most of the authors’ conclusions are supported by the findings with a few exceptions. The fact that the authors were able to collect enough cell material for LC-MS analysis is notable given that Ca. T. oneisti is otherwise unculturable. I have several comments, most are minor.
Title: “PBP4 is involved in cell division of the longitudinally dividing bacterium Candidatus Thiosymbion oneisti.” Throughout the manuscript (e.g., lines 19-22, 224-227, 384-387), the authors are circumspect about the role of PBP4TO in cell division but not in the title.
Figure 2a: The authors should consider labeling the domains of PBP4 with color and text to accommodate those readers with color blindness.
Figure 3b: Recommend writing “vector” and “pPBP4TO” along the vertical axis of the upper and lower micrographs, respectively.
Line 168-169: “The antibody specificity and recognition of PBP4TO were verified by western blot in Ca. T. oneisti protein extracts (Figure 3a).” Here, the authors should note the limitation of their results for the reader. While the E. coli PBP4 polyclonal antibody appears to bind PBP4TO, the signal is nevertheless weak. This could suggest that certain antibodies in the serum do not bind PBP4TO, which may be important for interpreting the results from the immunostaining in Figure 4.
Lines 181-182: “In constricting cells, PBP4TO localized at the new poles and at constriction sites (Figure 4).” The fluorescence micrographs in Figure 4 clearly show that PBP4 localizes to the poles in Ca. T. oneisti. However, whether PBP4 preferentially localizes to constriction sites is not clear. In Figure 4a, the fluorescence signal is quite diffuse in the constricting cell, whereas in Figure 4b we see evidence of PBP4TO at the leading edge of the division plane. The authors should consider comparing PBP4TO localization to FtsZTO (e.g., PMID: 29576473) to help clarify their microscopy in Figure 4. As it stands, it appears that PBP4 is weakly associated with the division plane.
Figure 5d: missing the “l” in “natural” in the left panel.
Lines 273-274: However, the overexpression of PBP4EC is toxic and results in cell lysis possibly because of a weakened wall [18].” There is a tendency in the literature to use words like “stressed” and “weakened” to describe defects in the cell wall structure. These terms are too vague. The authors should be more specific. PBP4 cleaves peptide crosslinks in the cell wall, which are required to maintain structural integrity.
Lines 277-279 and lines 284-286: The authors were unable to reproduce the toxicity normally associated with dacB overexpression. Have the authors considered overproducing the PBP4TOS69A variant (e.g., PMID: 20545860)?
Was there a reason the authors did not directly test PBP4TO for cell wall degrading activity? Since PBP4TO overproduction did not give the expected result, could this indicate that its activity departs from the authors’ expectation.
Other
Lines 30-32: Statement reads awkwardly.
Lines 105-108: Statement reads awkwardly.
Lines 99-100: Recommend moving this statement to the end of the paragraph to better introduce the Results section.
Line 233: E. coli (ital)
The manuscript should be edited for greater clarity.
Reviewer 2 Report
The manuscript by Wang and colleagues characterizes PBP4 from the gamma-proteobacterium Candidatus Thiosymbion oneisti. This is a particularly interesting organism because, contrary to most rods, it divides along the longer cell axis. Therefore, it is likely to have different mechanisms involved in cell division. However, the fact that Ca. T. oneisti is not amenable to genetic manipulation, considerably hinders cell division studies. This manuscript therefore resorts to immunofluorescence and in vitro experiments to characterize the endopeptidase PBP4.
Comments
Fig 4 – My main concern with this manuscript is related to PBP4 localization: it is very clear that in young cells PBP4 is localized at the cell poles. However, it is far less clear that as the septum closes, PBP4 stays at the leading edge of the septum, making it a candidate to be involved in new peptidoglycan synthesis during septum formation. In the second panel of Fig 4b, the cell is slightly invaginated, and there are two foci (top and bottom) that could be at the leading edge of the septum can be, but there is also a foci in the middle of the cell. In the third panel, where the septum is almost closed, there are foci at the poles and then there is a foci in the middle of the cell, but there are extra foci, so it is not clear if PBP4 specifically localizes at the leading edge of the septum. The side view does not help, as it seems a continuous string of foci. It would be helpful if the authors included a whole field of view in SI material, so that reader can assess the localization pattern in a large number of cells.
Also, Fig 4 would be easier to interpret if there was a clear definition of groups 1-4 (do they correspond to the schemes in panel g? If so, please write the “group 1, 2.. below each image). It seems that groups were not defined in a consistent manner (corresponding to a cell cycle stage) as there are 60% of cells in group 1 in panel d but only 32% in panel e. It is also not clear why each group includes cells of different sizes in the MreB and PBP4 analysis.
Still regarding Fig 4, the legend should mention the difference between panel a and b, as well as the 90º rotation of the 3D SIM images
Line 227 – to say that PBP4 signal overlaps PG incorporation sites, the authors would have to localize PBP4 in EDA-DA labelled cells. Otherwise, it is better to say for example that PBP4 localization is compatible with PG incorporation sites.
Fig 5, panel c – the Y axis is not a relative %, but relative fraction. Also, legend for panel d is missing (incorrectly assigned to panel c)
Fig 6 – Why is the first lane (no antibiotic) in the Penicillin G gel so much stronger than in the gels for other antibiotics? Also, can authors please double check the quantification in panel b? The drop between the first and second lane for PenG seems much larger than for Ampicillin, but in the graph it is identical. Are there other examples of in vitro assays where the effect seen for meropenem and aztreonam is seen for different PBPs?
Line 290 – Please rephrase, as authors do not have evidence that the membrane defects observed cause increased permeability
Figure 8 can be moved to SI and a Coomassie gel or any other loading control should be added.
Line 474 – How many cells (volume and OD, mg) were used for PG purification?
Minor comments
Fig 3c – “Fluorescence” in the Y axis has a typo.
Line 221 – last word should be localized, not localization.
Line 232, 233, italics for genes/species are missing
Fig 5 panel d – right gel – “natural” instead of “natura”
Fig 7 – A “4” is missing after PBP in the legend
Line 348 – “less” instead of les”
Line 350 – an “i” is missing in species name
Line 390 – last word “general” instead of “generally”
Reviewer 3 Report
This study by Wang et al. presents a study of a protein produced by the marine bacterium Candidatus Thiosymbion oneisti (TO) as a homolog of Escherichia coli penicillin-binding protein 4 (PBP4). They demonstrate that this putative PBP4 binds to the fluorescent beta-lactam Bocillin-FL and it localizes to the septum suggesting its function with cell division during longitudinal growth. Whereas the manuscript requires revision to correct grammar, and the study itself appears to have been conducted with care and attention to detail, there are a number of issues that need to be addressed to complete the study; at present, this work is considered preliminary.
Major issues:
- First and foremost, the authors base their identification of the recombinant protein based on sequence alignments of its encoding gene dacB to E. coli PBP4, while demonstrating a very different penicillin-binding profile between the two. Nonetheless, they claim the PBP to be an endopeptidase/carboxypeptidase based on this alignment, and discuss its likely involvement in the peptidoglycan metabolism of the bacterium. As this represents the first study on this particular protein, the authors need to demonstrate its enzymatic activity to confirm its hypothetical identity. Ideally, this should be done as in, eg., Clarke et al. (2009) Biochemistry 48(12):2675-2683. To assume the enzymatic activity and discuss it at length is inappropriate.
- At the outset of the study, the extent of peptidoglycan crosslinking is determined which the authors conclude is unusually high. However, as they then discuss (lines 318-321) the method used for quantification of muropeptides is inappropriate. Indeed, it was and the MS-based method used would not be able to provide “molar abundancies” as stated (line 502). If it is to be included in the manuscript, the extent of crosslinking needs to be determined using an appropriate methodology (which are based on Glauner’s original work ((1988) Anal Biochem 172(2) :451-464).
- The authors predict a molecular structure of “PBP4” and then present an extensive comparison of it to the known structure of E. coli PBP4 (lines 144-161). However, this comparison is inappropriate because the TO structure was predicted by threading its amino acid sequence on the known crystal structure of the E. coli PBP using PHYRE. Consequently, its structure will have to adopt that the one used for the prediction. All that can be said about this prediction is that the TO enzyme likely adopts this fold, and regions where the amino acid sequence does not can be identified and perhaps discussed.
Minor points:
- Lines 92-144: This section of the introduction is somewhat confusing as it speaks to both what is known about the E. coli PBP4 and summarizes the results of the current study. This section should be edited to first present what is known in the literature and then followed by the predictions for TO.
- Table 1 and line 502: Were “molar” percentages and values actually determined, and if so how. This would not be possible based on MS analysis alone.
- Line 136: should read “The genome encoded by Ca. T. oneisti hypothetical PBPs were identified…”
- Line 138: should read, “peptidase appears to be present in the …”
- Figure 2: The superimposed structures of (c) should be removed and the figure should, perhaps, highlight the differences between the two to highlight where the threading was not allowed by the Phyre program.
- Lines 237-244: Given that Bocillin-FL appears to bind to two distinct sites, it is not clear how the data presented here truly “identify” the catalytic residue. Whereas this identification of S69 is correct, either unique Bocillin binding or a kinetic activity analysis would be needed for this; ideally, the latter.
- Line 238 -9: should read “serine 69 was replaced with alanine as non-functional….although the S69A variant lost… (as amino acids are replaced, not mutated).
- Legend to Figure 5 needs additions/edits/corrections: Panel (b) does not present data for (S69A)PBP4TO; panel (c) Presents data for Bocillin binding which is not described in the legend; (d) and not (c) presents the native gel PAGE. With respect to the latter, the type of staining and how the samples were “denatured” should be indicated.
- Figure 5 (c): Unless the explanation is that the panel is mislabeled, it is not clear how/why there appears to be increased binding to (S69A)PBP4 of Bocillin as its concentration decreases. This needs to be addressed, if in fact the case.
- Line 284 and Figure 8: A possible explanation for the differences between function of PBP4To relative to PBP4Ec within the E. coli host could be improper localization of the PBP4TO to the outer leaflet of the inner membrane. In this regard, what fraction of the cell were the proteins analyzed for in Figure 8? Ideally, either the inner membrane fraction and/or the periplasmic content should be analyzed.
- Lines 347-348: It is not clear how the “smaller adaptation range” of the cell may influence peptidoglycan metabolism of this Gram-negative bacterium given this cell wall layer primarily, if not solely, holds the cell together rather than protects from exogenous agents. This section of the Discussion warrants further explanation if to be retained.
- Line 356-357: To justify this speculation for PBP4To as representing the dominant endopeptidase in the bacterium, minimally the protein needs to be clearly shown to function as endopeptidase.
- Lines 417-421: An expansion of the cloning of the PBP4 gene should be provided – from Table 2, presumably dacB.
Round 2
Reviewer 1 Report
No further comments.
Author Response
We thank the reviewer for their careful reading of the manuscript and their constructive remarks.
Reviewer 3 Report
The authors appear to have missed the point about the use of the Phyre predicted model. Any comparison to the Ec PBP4 structure is meaningless, because the predicted structure is thread onto it - clearly has to be the same - Except where apparent differences exist which are evident from the lack of predicted structure such as the dashed lines in the beta sheet Domain III - the fact that the structure was not complete here indicates some difference. And so again, the superimposition of the two structures is both meaningless and misleading.
Also, the authors appear to agree that using the integration of MS data is not appropriate to determine absolute molar concentrations and then turn around and argue for its application. Indeed, MS data can be used carefully for comparisons of similar samples, but without being able to determine absolute molar coefficients using appropriate standards, any quantification is an approximation and should be recognized as such.
Author Response
We thank the reviewer for their careful reading of the manuscript and their constructive remarks.
Point 1. The authors appear to have missed the point about the use of the Phyre predicted model. Any comparison to the Ec PBP4 structure is meaningless, because the predicted structure is thread onto it - clearly has to be the same - Except where apparent differences exist which are evident from the lack of predicted structure such as the dashed lines in the beta sheet Domain III - the fact that the structure was not complete here indicates some difference. And so again, the superimposition of the two structures is both meaningless and misleading.
We deleted superimposition image according to recommendation. The PBP4TO model is created by Phyre based on its sequence similarity to PBP4EC. As a homologue of PBP4EC, it also makes sense that PBP4TO domain I and II show similar structures as the corresponding domains of PBP4EC, especially for the conserved transpeptidase/penicillin-binding (PD) domain I for all PBPs. As it is not our intention to claim that the structure of PBPTO is correct, which is mentioned in the manuscript, we have deleted the overlay, although we think that it was useful as it showed that the two structures have differences.
Point 2. Also, the authors appear to agree that using the integration of MS data is not appropriate to determine absolute molar concentrations and then turn around and argue for its application. Indeed, MS data can be used carefully for comparisons of similar samples, but without being able to determine absolute molar coefficients using appropriate standards, any quantification is an approximation and should be recognized as such.
The material used for PG purification was collected from around 3000 nematodes. As an uncultivable bacterium, the Ca. T. oneisti cells collected from environment are very limited and not enough to be measured by the UV-VIS spectra quantification method. We agree that the quantification of MS data is an approximation, but we can only perform MS analysis because of the material limitation. The cross-linkage level of different species can still be compared to some extent. To make it clarified, we rephrase the discussion in the manuscript as “Hence, the result might not be completely comparable to previous data obtained from absorbance analysis at 204 nm on which most published data on the PG composition of common model organisms are based. However, using this method to analyze Vibrio cholerae’s PG [51] shows a degree of crosslinkage of 27.37% comparable to the 24.49% that was reported in [52] by the UV absorbance profile at 204 nm.” (Line 322-326)